# Should Firms in Emerging Markets Invest in R&D? Evidence from China's Manufacturing Sector

Dachen Sheng [1,2,*] and Heather Montgomery [1]

1   Department of Business and Economics, International Christian University, 3-10-2 Osawa, Mitaka 181-8585, Tokyo, Japan
2   International College of Liberal Arts, Yamanashi Gakuin University, 2-4-5 Sakaori, Kofu 400-8575, Yamanashi, Japan
*   Correspondence: g239009s@icu.ac.jp

**Abstract:** Analyzing micro-level firm data from the Chinese manufacturing sector, this study provides compelling evidence that firms in emerging markets that invest in research and development (R&D) for product differentiation significantly increase firm performance as measured by market power, profitability, and earning quality. Privately held (non-state-owned), mid-size, Shenzhen exchange-listed firms experience the largest boost to firm performance when they invest in research and development. However, analyzing the contribution of R&D investment to firm market value, we reveal that while R&D investments are valued by institutional investors, the potential for investment in R&D to boost firm performance is not recognized by individual investors, who dominate Chinese financial markets. This finding suggests that managers may under-invest in R&D if equity compensation comprises a large share of the overall compensation package.

**Keywords:** R&D; product differentiation; market power; profitability; earning quality

## 1. Introduction

Product differentiation is an important management strategy for coping with competition. As far back as Smith (1956), it has been recognized that differentiation has the potential to match goods and services more efficiently to clients at minimum cost, avoiding the waste inherent in producing goods and services for which there is little or no demand. In fact, lack of differentiation, the threat of substitute services or products, is identified as one of Porter's (1979) famed five competitive forces that shape management strategy.

Concrete benefits accrue to firms that successfully differentiate their products and services. Hermalin and Katz (2013) illustrate that product differentiation allows producers to realize economic rents. These rents accrue through price differentiation (Ekelund 1970) or market power (Perloff and Salop 1985). Firms with widely differentiated products are often large, serving many clients (Yin and Zuscovitch 1998), and the resulting dominant market position may help defend the firms from the threat of potential new entrants into the market. Therefore, the larger market share arising from successful differentiation yields market power (Rhoades 1985) and higher profits (Başgoze and Sayin 2013).

The benefits of more market power and higher profits that potentially accrue to firms with differentiated products and services are of course enticing to managers. Brown et al. (2009) has documented that it is indeed these potential financial gains from R&D that incentivize firms to invest in it. However, service and product differentiation require investment in research and development (R&D) to support the creation of new products with differentiated features. Product differentiation research is rarely a minor update of technology; it almost always entails research into processes themselves, requiring major technological innovation. Due to these barriers, large firms in highly competitive markets demonstrate higher rates of investment in R&D and higher product differentiation (Cellini and Lambertini 2002). Consistent with this stylized fact, De Meyer and Mizushima (1989)

report that localization of competition increases the level of R&D investments and Lin and Saggi (2002) find that more competitive Bertrand firms usually exhibit more product differentiation.

Other benefits apparently accrue to these firms as well: Hombert and Matray (2018) finds that US firms with larger R&D investment perform better in product differentiation and are less affected by the negative sides of globalization, such as large imports from competing markets such as China.

Despite these documented advantages, investment in R&D is not a foregone conclusion. The expenditure required for research and development of product differentiation is significant and returns on such an investment are uncertain: research can and does fail. Analyzing Vietnamese firm data, Ninh et al. (2018) show that firms' risk aversion is the main factor dissuading firms from investing in R&D. These disincentives for R&D investment may be particularly strong in developing countries.

Indeed, the inception of new process research usually occurs in developed countries. Although Govindarajan and Ramamurti (2011) have documented some rare examples of what is referred to in the literature as reverse innovation, in general, because R&D related to product differentiation usually requires major technological innovation, new process research is birthed in developed economies which enjoy technological advantages. The new processes may then be implemented in actual production and diffused in emerging markets where labor costs are lower. However, the established natural order of product differentiation flows from technologically advanced economies to low labor cost emerging markets. This study challenges this established order of innovation, investigating whether the benefits accruing to firms with differentiated services and products outweigh the costs of the R&D required to differentiate goods and services in emerging markets as well.

Its unique characteristics as "the world's factory" make the Chinese market an ideal laboratory in which to examine this question of whether firms in emerging markets are best served by the existing status quo or whether the benefits accruing from product differentiation make investments into R&D a better management strategy. China's phenomenal economic success story started in the late 1990s thanks to an economic strategy focusing on exports and foreign direct investment. In the decade between 2000 and 2010, China made significant investments in developing infrastructure and providing subsidies and tax benefits to exporting firms. Growth remained robust in the following decade as well with China's decision to join the World Trade Organization. The most recent decade has also seen a healthy rebalancing of the Chinese economy. The recent rebalancing and resulting increase in domestic demand provides Chinese manufacturers a new market that may warrant new marketing strategies. Chinese firms face a choice: continue to focus on overseas markets, exporting and taking advantage of government tax subsidies, or invest in R&D, differentiate their products, and compete in the domestic Chinese market. Managers choosing the latter strategy are those that calculate the potential returns from expanding into the local Chinese markets outweigh those of continuing to focus on overseas markets, even after factoring in the costs of the R&D required for a local market diversification strategy.

Against this background, the purpose of this study is to investigate whether conducting research and innovation to realize product differentiation in the Chinese market has the potential to improve the performance of Chinese firms. This study contributes to the existing research in several ways. First, we confirm that despite its position in the global economy as an emerging market, the gains from product differentiation are significant in the Chinese market. Next, we explore whether the improvements in firm performance resulting from R&D investment translate into higher returns for shareholders. Thirdly, we take advantage of China's unique institutional setting to analyze whether the benefits of R&D for product differentiation differ by listing location and ownership: state-owned vs. private firms. Finally, we test for potential heterogeneity in the market reaction of different classes of investors to firms' R&D investments.

The rest of the paper is organized as follows. The next section surveys the existing literature, developing several hypotheses to be investigated. Section 3 describes the data and the research methodology used in this study. Section 4 reports and interprets the regression results. Finally, Section 5 concludes and provides some policy suggestions.

## 2. Literature Review and Hypotheses

Based on a survey of the existing body of research on the role of R&D investment in management strategy, we formulate six distinct hypotheses related to the impact of R&D on firm performance. The first three hypotheses hone in on the core benefits accruing to firms that invest in R&D: market power, profitability, and earnings quality. If these hypotheses hold, then we also expect shareholders of firms that invest in R&D to enjoy higher shareholder benefits. The final three hypotheses explore potential heterogeneity in the effect of R&D on those firm advantages: listing location and ownership structure— whether firms are state-owned or private and, for private firms, whether shareholders are individual investors or institutional investors.

Drawing on Porter's (1979) "five forces" analysis, our first hypothesis centers on the relationship between investment in R&D and market share in the Chinese market. Readers are presumably already familiar with Porter's (1979) now-famous five competitive forces, beginning with the threat of new entrants, that Porter (1979) contends determine the state of competition in an industry and should therefore shape the firm's business strategy. In a more mathematical model of the relationship between the degree of competitiveness in a market and R&D expenditure, Matsumura et al. (2013) suggest that R&D activities intensify when the market has very weak or very intense competition, but R&D incentives weaken when competition is moderate. Further, in a Cournot model of market competition, Ishida et al. (2011) suggest an increase in the number of inefficient, high-cost firms in a market can stimulate R&D by efficient, low-cost firms.

**H1:** *Firms with higher R&D investment relative to other firms and across time control relatively higher market share.*

Next, we turn to the impact of R&D investment on profitability. Although some researchers have suggested that there may be a threshold level of R&D beyond which R&D investment contributes little to firm performance measures (Chen and Ibhagui 2019), there is evidence that firms with higher R&D perform better operationally (Eberhart et al. 2004) and are more productive (Belderbos et al. 2004). The pattern of technology adoption in emerging markets suggests we may expect small but persistent technological improvements and updates of products and services. Such performance enhancements should be reflected in short-term future firm performance.

**H2:** *Firms with higher R&D investment relative to other firms and across time enjoy relatively higher profits as measured by ROA.*

In addition to the effect of R&D investment on overall profitability, we investigate the impact of R&D expenditures on earnings quality. Franzen and Radhakrishnan (2009) show that for profitable firms, R&D has a positive impact on future earnings. Aharony et al. (2010) find that while temporary changes in investment are not a good indicator of future earnings, overall, growth in investment forecasts future returns.

**H3:** *Firms with higher R&D investment relative to other firms and across time report higher earnings quality as measured by net profit margins.*

If hypotheses H1, H2, and H3 about the positive impact of R&D investments on market power, profitability, and earnings quality are correct, then we would expect shareholders of firms with higher levels of R&D investments to benefit in terms of higher valuation of their shares. In developed economies, R&D investment has been documented to have a significant and long-term valuation impact (Sougiannis 1994). Griliches (1981) uncovers a significant relationship between 'intangible' capital such as R&D expenditures and firm

market value. There is a significant positive relation between Tobin's Q and announcements of increases in R&D expenditures (Szewczyk et al. 1996).

**H4:** *Firms with higher R&D investment relative to other firms and across time post higher share valuation.*

The two main stock exchanges in China, the Shanghai and Shenzhen exchanges, differ in their salient characteristics. Firms listed on the Shanghai stock exchange tend to be larger, more established firms. Firms in the high-tech industry tend to gravitate to the Shenzhen stock exchange, which is more growth-oriented and includes smaller firms. The inherent nature of high-tech firms, which tend to list on the Shenzhen exchange, may make those firms more likely to invest in R&D. Indeed, existing research suggests that the different characteristics across the two exchanges may yield heterogeneity across the effects of R&D on firms' outcomes. For example, Tsai and Wang (2005) find that firms on both ends of the spectrum in terms of asset size—smaller and larger firms—are more willing to invest in R&D. This result holds across both high-tech and traditional sectors.

**H5:** *Shenzhen exchange-listed firms are more likely to invest in R&D.*

One well-known characteristic of Chinese capitalism is the prevalence of state-owned enterprises. State-owned firms function as business entities, so optimize profitability, but also have a second bottom line of meeting the social or political objectives of the government (Choe and Yin 2000). There is evidence that pursuing these alternative objectives increases employees' loyalty and recognition (Zhu et al. 2014). Indeed, there is empirical evidence that state ownership can be financially beneficial for firms (Zhu et al. 2016). We expect that the broader set of objectives pursued by state-owned firms may result in more corporate social responsibility for those firms, which has been shown to positively influence sustainability governance and sustainable performance (Khan and Ghouri 2022). Therefore, as illustrated by Azzam and Alhababsah (2022), state ownership may increase the likelihood that firms invest in R&D. At the same time, however, the pursuit of the double bottom line of profitability and social responsibility constrain firms' decision-making (Liu and Zhang 2017). Perhaps, for this reason, state-owned firms in China seem to be less efficient in transferring R&D into final output (Zhou et al. 2017).

**H6:** *Privately owned firms enjoy more economic benefits from R&D.*

Another distinctive feature of China's financial markets is that in China, individual investors tend to be more active in financial markets. Financial markets in developed economies such as the US tend to be dominated by institutional investors such as mutual funds. The behavior of institutional investors can be quite different from that of individual investors (Ng and Wu 2007). Institutional investors are likely to investigate firm fundamentals and the economic environment in which a firm is operating when making investment decisions. Individual investors have less access to information and analysis, leading their investment decisions to be myopic and focused on readily available information such as earning per share ratios. There is evidence that individual investors tend to herd and form investment consensus (Ng and Wu 2010), affecting price momentum (Chui et al. 2010). Thus, based on the existing body of knowledge, we expect this characteristic of a higher percentage of individual investors in China's financial markets to influence firm decisions on undertaking R&D. This will be reflected in a higher share of institutional investors ownership at firms that undertake relatively more R&D.

**H7:** *Firms with higher R&D investment relative to other firms and across time yield a higher share of institutional investor ownership, as measured by the share of mutual fund investors.*

### 3. Data and Empirical Methodology

*3.1. Data*

The data used in the analysis is the full universe of manufacturing firms listed on either the Shanghai or Shenzhen exchanges in China as of 2017 as reported in the Choice database. After cleaning the dataset by removing firms experiencing financial distress, as defined by reporting a loss for two consecutive years, the final dataset includes detailed annual financial statements on over 1264 listed firms from 29 sub-industries within the manufacturing sector over the 5-year period 2017–2021, for a total sample of 5056 observations for most data series. Descriptive statistics and variable definitions are provided in Tables 1 and 2, respectively.

**Table 1.** Descriptive statistics.

| Variable | Unit | Observation | Mean | Standard Deviation | PCTL (25%) | PCTL (75%) |
|---|---|---|---|---|---|---|
| Market Share | Percentage | 5056 | 2.294 | 5.747 | 0.193 | 1.729 |
| NETMARG | Percentage | 5055 | 0.148 | 147.283 | 1.844 | 11.081 |
| ROA | Percentage | 5056 | 5.081 | 9.778 | 2.433 | 8.599 |
| Tobin's Q | Percentage | 5056 | 3.043 | 5.614 | 1.347 | 3.202 |
| INST | Percentage | 5056 | 2.970 | 5.039 | 0.051 | 3.478 |
| RDRATIO | Percentage | 5056 | 2.180 | 1.720 | 1.002 | 2.957 |
| TOP | Percentage | 5056 | 32.388 | 13.907 | 21.830 | 40.980 |
| LIAB | Percentage | 5056 | 43.738 | 32.767 | 28.571 | 56.684 |
| DUAL | Binary | 5056 | 0.242 | 0.428 | 0 | 0 |
| SHORTLIAB | Percentage | 5056 | 83.487 | 15.014 | 75.689 | 95.239 |
| SOE | Binary | 5056 | 0.336 | 0.472 | 0 | 1 |

**Table 2.** Variable definitions.

| Variable | Symbol | Variable Treatment |
|---|---|---|
| Market Share | Market Share | Revenue of firm/Sum of the total revenue of all firms in the industry |
| Net profit margin | NETMARG | Net profit/Revenue |
| Return on Assets | ROA | Net profit/Total assets |
| Tobin's Q | Tobin's Q | Firm market value/Equity value |
| Mutual fund ownership percentage | INST | Shares own by mutual funds/Total number of shares issued |
| Research and Development expense to asset ratio | RDRATIO | R&D expense/Total assets |
| Top shareholder's ownership | TOP | Ownership of largest shareholder/Total shares |
| Leverage ratio | LIAB | Liability/Total assets |
| The Board chairman and CEO are the same person | DUAL | Binary, if the board chairman is also CEO, DUAL = 1, otherwise = 0. |
| Short-term liability percentage | SHORTLIAB | Current debt/Total debt |
| State-owned enterprise | SOE | Binary, if there are non-private sector shareholders, SOE = 1, otherwise = 0. |

The market share of each firm is calculated as the firms' revenue expressed as a percentage of the total industry revenue. Firms with any shareholders other than private-sector shareholders are classified as a state-owned firm. The other variables reported in Tables 1 and 2 are from firm financial reports from the Choice database.

The descriptive statistics reported in Table 1 show that the average ROA is a little above 5 percent, reflecting the average return on assets in the manufacturing industry. The R&D to asset ratio has a mean of 2.18 percent. This is roughly the middle point between the 25 and 75 percentiles for the R&D to asset ratio, suggesting that the research and development expense to asset ratio for the firms in the sample are normally distributed. The mean of the variable DUAL—a dummy variable that takes the value of 1 if the chairman of the board

of directors also serves as the CEO of the firm, reflects the fact that about a quarter of the listed manufacturing firms in China have the same person in the role of manager and chair of the board of directors. This concentration of power presents advantages when decisions need to be taken quickly. However, when making an investment decision on R&D, the risk of failure is high if the person with such a high concentration of power is not a specialist or experienced professional. Another key variable, SOE, is a dummy variable that takes the value of one if a firm has at least one state shareholder. As evidenced by the mean value in the summary statistics, about one-third of the manufacturing firms listed in China have state ownership.

### 3.2. Empirical Methodology

#### 3.2.1. Baseline Model: Short-Run Firm Performance

Equation (1) represents the baseline model testing the relationship between R&D and the market power as expressed in the first hypothesis (H1).

$$
\begin{aligned}
Market\ Share_{i,t+1} = {} & \beta_0 + \beta_1 RDRATIO_{i,t} + \beta_2 TOP_{i,t} + \beta_3 LIAB_{i,t} + \beta_4 DUAL_{i,t} + \\
& \beta_5 SHORTLIAB_{i,t} + \beta_6 (RDRATIO_{i,t} * TOP_{i,t}) + \varepsilon_{i,t+1}
\end{aligned}
\tag{1}
$$

Parallel to the baseline regression for estimating the impact of R&D on firm market share, Equations (2) and (3) estimate the effect of R&D on firm profits, the second hypothesis (H2), as measured by ROA, and earnings quality, the third hypothesis (H3):

$$
\begin{aligned}
ROA_{i,t+1} = {} & \beta_0 + \beta_1 RDRATIO_{i,t} + \beta_2 TOP_{i,t} + \beta_3 LIAB_{i,t} + \beta_4 DUAL_{i,t} + \\
& \beta_5 SHORTLIAB_{i,t} + \beta_6 (RDRATIO_{i,t} * TOP_{i,t}) + \varepsilon_{i,t+1}
\end{aligned}
\tag{2}
$$

$$
\begin{aligned}
NETMARG_{i,t+1} = {} & \beta_0 + \beta_1 RDRATIO_{i,t} + \beta_2 TOP_{i,t} + \beta_3 LIAB_{i,t} + \beta_4 DUAL_{i,t} + \\
& \beta_5 SHORTLIAB_{i,t} + \beta_6 (RDRATIO_{i,t} * TOP_{i,t}) + \varepsilon_{i,t+1}
\end{aligned}
\tag{3}
$$

Equations (1)–(3) above estimate the impact of investment in R&D on firm market share, profitability, and earnings quality, after controlling for various firm-level characteristics as defined above in Table 2.

In each equation, $\varepsilon_{i,t+1}$ is a rational expectations error term, which is serially uncorrelated orthogonal to information available at time t. The expectation conditional on time t information, $I_t$, is $[E[\varepsilon_{i,t+1}|\ I_t] = 0$, so that period-t instruments are valid.

One time period may seem too short to fully capture the effects of R&D on firm outcomes such as market share, profits, and earnings quality, even when estimation is carried out with annual data. The Chinese manufacturing sector, however, is characterized by frequent technology updating. As in many emerging markets, manufacturing firms in China continuously adopt technologies from developed markets, using the comparative advantage of low-cost labor to utilize new technology in production. Even in their role as technological adopters or followers, or perhaps precisely because of their position in playing catch-up to the cutting-edge technology in developed markets, the pace of technological improvement in emerging markets is usually faster than in developed economies. Thus, for our baseline estimates, we assume that the effects of R&D are enjoyed by firms in the following period. This issue will be further explored below.

While the baseline estimate of R&D on firm outcomes is expected to be positive and statistically significant, the interaction term of R&D and "TOP", the variable for largest shareholder's holding, examines whether centralized ownership power perhaps constrains the positive impact of R&D on firm market power.

#### 3.2.2. Robustness Check: Longer-Run Analysis

As argued above, there are compelling reasons to think that the effects of R&D on firm performance are enjoyed by firms in emerging markets even over a relatively short time period. Perhaps especially in China, fast adoption of foreign technology is recognized as one of the drivers of growth and is often one of the features highlighted in analysis

of China's phenomenal economic growth, sustained over the longest period in economic history. At the same time, it is recognized that the speed of technological adoption and the growth enjoyed as a result of technological progress usually slows dramatically as emerging market economies approach the technological frontier.

Therefore, we expand on our baseline analysis of Equation (2) by investigating the influence of R&D on firm profits in the current year and two years in the future.

### 3.2.3. Shareholder Interests

If our hypothesis that R&D positively, significantly impacts firm market share, profits and earnings quality is true, then we would also expect firm shareholders to enjoy higher share prices and returns. However, this is not a foregone conclusion. Agency theory highlights how management's behavior may deviate from shareholders' interests to maximize their own interests. If managers think R&D is a "good gamble", they may invest in R&D regardless of the risk appetite of shareholders. The reverse is also true. If firm shares make up a large part of management compensation packages, for example, managers may be more risk averse, failing to pursue research and development projects that are risky, yes important to the firm's continued development.

In Equation (4), Tobin's Q is included as a dependent variable, enabling us to estimate the impact of R&D expenditures on an overall market price indicator, giving an estimate of the effects of R&D on shareholders:

$$
\begin{aligned}
Tobin's\ Q_{i,t+1} =\ &\beta_0 + \beta_1 RDRATIO_{i,t} + \beta_2 TOP_{i,t} + \beta_3 LIAB_{i,t} + \beta_4 DUAL_{i,t} + \\
&\beta_5 SHORTLIAB_{i,t} + \beta_6 (RDRATIO_{i,t} * TOP_{i,t}) + \varepsilon_{i,t+1}
\end{aligned}
\tag{4}
$$

### 3.2.4. Heterogeneity across Firms: Listing Location and Ownership Model

Next, we turn to explore two sources of possible heterogeneity in our baseline result. First, we divide the sample into two groups according to the exchange where the firms chose to list. The firms listed in Shanghai are larger in asset size and are more traditional manufacturing biased. On the other hand, the firms listed in Shenzhen tend to be smaller, more high-tech, and typically more heavily invested in R&D.

Next, we analyze whether the interaction between R&D and firm performance outcomes vary between private companies and state-owned enterprises. State-owned firms may be reluctant to invest in research and development; choosing lower, more stable returns over the more volatile, but potentially much higher, returns from R&D. Management at state-owned firms may be loss-averse; the utility cost of possible failure of R&D investments outweighing the benefits enjoyed in the case of success.

Finally, we investigate one other salient feature of Chinese financial markets: the dominance of individual investors over institutional investors. Our hypothesis is that individual investors experience more information asymmetries than institutional investors. Institutional investors such as mutual funds often have professional analysts covering various industries, who conduct onsite firm visits and have detailed knowledge of the firms' production system and efficiency. Individual investors are limited to publicly available information and may rely on rule-of-thumb analysis such as the firm's earnings per share. Firms investing in R&D may have costs today which lower their earnings, but enjoy high earnings in the future, when the returns from R&D accrue to the firm. If individual investors focus only on current earnings, they may fail to recognize the full potential of firms that invest heavily in R&D.

To explore these issues further, we estimate Equations (1)–(3) on subsample of firms listed on the Shanghai and Shenzhen exchanges as a test of hypothesis five, H5. To test hypothesis six, H6, we re-estimate Equations (1)–(3) on subsamples of state-owned enterprises and privately held forms. Finally, we estimate Equation (5) below, in which the

effect of R&D on mutual fund ownership, serving as an indicator of institutional investor interest in the firm, is estimated:

$$INST_{i,t+1} = \beta_0 + \beta_1 RDRATIO_{i,t} + \beta_2 TOP_{i,t} + \beta_3 LIAB_{i,t} + \beta_4 DUAL_{i,t} + \\ \beta_5 SHORTLIAB_{i,t} + \beta_6(RDRATIO_{i,t} * TOP_{i,t}) + \varepsilon_{i,t+1} \quad (5)$$

## 4. Results and Discussion

### 4.1. Baseline Results

Table 3, columns 1 and 2 show the results of empirical estimation of Equation (1). After controlling for industry-specific and year-specific idiosyncratic changes in market shares, the estimated coefficient on the R&D to asset ratio is positive and statistically significant, confirming hypothesis one, H1. Higher R&D investment leads to higher market power in the following year. The interaction term between the R&D ratio and the ownership concentration ("TOP") has a negative and statistically significant coefficient estimate. This supports the hypothesis laid out above that more centralized ownership—common in state-owned firms—reduces the positive effects of R&D investment on market power.

**Table 3.** Research expense, market power, profitability and earning quality.

| | Dependent Variable | | | | | |
| | Market Shares$_{t+1}$ | | ROA$_{t+1}$ | | NETMARG$_{t+1}$ | |
| | (1) | (2) | (3) | (4) | (5) | (6) |
|---|---|---|---|---|---|---|
| RDRATIO | 0.127 | 0.401 *** | 0.789 *** | 0.782 *** | 6.369 ** | 6.445 ** |
| | (0.121) | (0.103) | (0.212) | (0.212) | (3.140) | (3.141) |
| TOP | 0.077 *** | 0.069 *** | 0.117 *** | 0.118 *** | 0.644 *** | 0.644 *** |
| | (0.010) | (0.008) | (0.017) | (0.017) | (0.247) | (0.247) |
| LIAB | 0.011 *** | 0.013 *** | −0.027 *** | −0.027 *** | −0.148 ** | −0.148 ** |
| | (0.002) | (0.002) | (0.004) | (0.004) | (0.063) | (0.063) |
| DUAL | 0.396 ** | 0.150 | −0.747 ** | −0.742 ** | 1.167 | 1.151 |
| | (0.187) | (0.157) | (0.327) | (0.327) | (4.844) | (4.843) |
| SHORTLIAB | −0.003 | −0.006 | −0.005 | −0.004 | 0.237 * | 0.232 * |
| | (0.005) | (0.005) | (0.009) | (0.009) | (0.139) | (0.139) |
| RDRATIO*TOP | −0.013 ** | −0.011 *** | −0.013 ** | −0.013 ** | −0.140 | −0.140 |
| | (0.003) | (0.003) | (0.006) | (0.006) | (0.090) | (0.090) |
| Constant | 0.141 | 0.990 | 2.180 ** | 1.933 * | −38.289 *** | −36.098 ** |
| | (0.562) | (0.666) | (0.984) | (1.017) | (14.591) | (15.081) |
| Industry Control within Manufacturing Sector | N | Y | N | N | N | N |
| Year Control | N | Y | N | Y | N | Y |
| Observations | 5056 | 5056 | 5056 | 5056 | 5055 | 5055 |
| R$^2$ | 0.029 | 0.331 | 0.030 | 0.032 | 0.004 | 0.005 |
| Adjusted R$^2$ | 0.027 | 0.326 | 0.029 | 0.030 | 0.003 | 0.003 |
| Residual Std. Error | 5.668 (df = 5049) | 4.718 (df = 5018) | 9.924 (df = 5049) | 9.914 (df = 5046) | 147.072 (df = 5048) | 147.086 (df = 5045) |
| F Statistic | 24.701 *** (df = 6; 5049) | 67.123 *** (df = 37; 5018) | 25.746 *** (df = 6; 5049) | 18.633 *** (df = 9; 5046) | 3.418 *** (df = 6; 5048) | 2.773 *** (df = 9; 5045) |

Note: ***, **, and * denote the statistical significance at the 1%, 5%, and 10%, standard errors are shown in parentheses.

Estimates of the relationship between R&D and firm profitability are reported in Table 3, columns 3 and 4. The estimated coefficients on R&D are similar to the previous estimates for market power: the R&D ratio has a positive and statistically significant coefficient estimate, indicating that R&D investment leads to higher profitability in the following period. Hypothesis two, H2, is also confirmed. In columns 3 and 4 as well, the interaction term of R&D expenditure and power concentration has a negative and statistically significant coefficient estimate. When firms need research and development to implement product differentiation, it seems that a concentrated management system is inefficient.

In Table 3, columns 5 and 6, the analysis turns to earnings quality. Columns 5 and 6 report the results of estimation of Equation (3), which estimates the relationship between R&D investment earnings quality as measured by net profit margins. The positive and

highly statistically significant coefficient estimates on R&D confirm hypothesis three, H3. The interaction term of R&D and power concentration is no longer statistically significant, suggesting that concentration of power may not significantly constrain the profit margin gains from R&D investment.

### 4.2. Robustness Check: Longer-Run Results

Table 4 reports the results of longer-term analysis. In particular, columns 1 and 2 report estimates of the relationship between R&D expenditure and ROA two years into the future. As a robustness check, the effect of R&D on contemporaneous ROA is also reported. The coefficients of R&D for both current and two years future ROA are positive and statistically significant. Taking the results reported in Tables 3 and 4 of analysis of the effect of R&D on current ROA, one-year ahead ROA, and two-years ahead ROA, all together, the coefficient of R&D for one-year ROA has the biggest marginal effect. Moreover, the coefficient of R&D for the one-year ROA is the most precise: it has the lowest standard error and highest statistical significance. Taken together, these findings confirm our speculation that in emerging markets R&D seeks product differentiation in order to influence short-term firm performance outcomes.

**Table 4.** Robustness check: Profitability.

| | Dependent Variable | | | |
| | $ROA_{t+2}$ | | ROA | |
| | **(1)** | **(2)** | **(3)** | **(4)** |
|---|---|---|---|---|
| RDRATIO | 0.701 *** | 0.696 *** | 0.352 * | 0.369 ** |
| | (0.236) | (0.236) | (0.184) | (0.184) |
| TOP | 0.104 *** | 0.106 *** | 0.100 *** | 0.099 *** |
| | (0.019) | (0.019) | (0.014) | (0.014) |
| LIAB | −0.020 *** | −0.020 *** | −0.133 *** | −0.133 *** |
| | (0.004) | (0.004) | (0.004) | (0.004) |
| DUAL | −0.923 ** | −0.917 ** | −0.677 ** | −0.681 ** |
| | (0.371) | (0.371) | (0.284) | (0.284) |
| SHORTLIAB | −0.006 | −0.005 | −0.006 | −0.007 |
| | (0.011) | (0.011) | (0.008) | (0.008) |
| RDRATIO*TOP | −0.011 | −0.011 | −0.003 | −0.003 |
| | (0.007) | (0.007) | (0.005) | (0.005) |
| Constant | 2.503 ** | 1.774 | 7.777 *** | 8.647 *** |
| | (1.117) | (1.148) | (0.856) | (0.884) |
| Year Control | N | Y | N | Y |
| Observations | 3792 | 3792 | 5056 | 5056 |
| $R^2$ | 0.025 | 0.028 | 0.221 | 0.224 |
| Adjusted $R^2$ | 0.023 | 0.026 | 0.221 | 0.223 |
| Residual Std. Error | 9.762 (df = 3785) | 9.748 (df = 3783) | 8.633 (df = 5049) | 8.620 (df = 5046) |
| F Statistic | 16.069 *** (df = 6; 3785) | 13.702 *** (df = 8; 3783) | 239.324 *** (df = 6; 5049) | 162.071 *** (df = 9; 5046) |

Note: ***, **, and * denote the statistical significance at the 1%, 5%, and 10%, standard errors are shown in parentheses.

### 4.3. Shareholder Interests

Thus far, our empirical analysis shows strong, robust benefits in firm performance from investment in R&D. The next step in our analysis takes up the question of whether firm investment in R&D translates into higher returns for investors. Using Tobin's Q as an indicator of the overall value of firm equity, we first estimate the relationship between firm investment in R&D and firm value for the entire sample. The results are reported in Table 5.

The results refute hypothesis four, H4, and thus at first, seem puzzling. We find no statistically significant impact of R&D investment on long-term (2-years out) firm value, and the coefficient estimates on R&D investment for short-term (the following year) firm value are statistically significantly negative.

**Table 5.** Research expense and Tobin's Q.

| | Dependent Variable | | | |
| --- | --- | --- | --- | --- |
| | Tobin's Q$_{t+1}$ | | Tobin's Q$_{t+2}$ | |
| | (1) | (2) | (3) | (4) |
| RDRATIO | −0.235 ** | −0.250 ** | −0.149 | −0.152 |
| | (0.117) | (0.117) | (0.130) | (0.130) |
| TOP | −0.028 *** | −0.026 *** | −0.019 * | −0.018 * |
| LIAB | 0.032 *** | 0.032 *** | 0.033 *** | 0.033 *** |
| | (0.002) | (0.002) | (0.002) | (0.002) |
| DUAL | 0.781 *** | 0.786 *** | 0.752 *** | 0.755 *** |
| | (0.181) | (0.181) | (0.205) | (0.204) |
| SHORTLIAB | 0.017 *** | 0.018 *** | 0.020 *** | 0.021 *** |
| | (0.005) | (0.005) | (0.005) | (0.006) |
| RDRATIO*TOP | 0.007 ** | 0.007 ** | 0.005 | 0.005 |
| | (0.003) | (0.003) | (0.004) | (0.004) |
| Constant | 0.930 * | 0.192 | 0.561 | 0.076 |
| | (0.546) | (0.562) | (0.615) | (0.632) |
| Year Control | N | Y | N | Y |
| Observations | 5056 | 5056 | 3792 | 3792 |
| R$^2$ | 0.040 | 0.047 | 0.052 | 0.056 |
| Adjusted R$^2$ | 0.039 | 0.046 | 0.050 | 0.054 |
| Residual Std. Error | 5.503 (df = 5049) | 5.484 (df = 5046) | 5.376 (df = 3785) | 5.366 (df = 3783) |
| F Statistic | 35.389 *** (df = 6; 5049) | 27.907 *** (df = 9; 5046) | 34.484 *** (df = 6; 3785) | 27.918 *** (df = 8; 3783) |

Note: ***, **, and * denote the statistical significance at the 1%, 5%, and 10%, standard errors are shown in parentheses.

However, even in developed markets, market participants underestimate the benefits of R&D, leading R&D being a major contributor to information asymmetry and insider gains (Aboody and Lev 2000). Gu (2016) posits that firms with higher levels of R&D investment are riskier but expected to have higher share returns, and empirical evidence largely supports the model's predictions. Firms with larger R&D usually have higher market share return volatility (Chan et al. 2001). Franzen et al. (2007) document that higher R&D spending increases the likelihood of misclassifying solvent firms as being in financial distress.

Our interpretation of the results in Table 5 is that investors focus on current earnings in making investment decisions, so there is a negative effect of R&D investment—which can cause a significant hit to earnings—on shareholders in the following period. However, the negative impact of R&D investment on shareholder value dissipates after the first period.

### 4.4. Heterogeneity across Firms: Listing Location and Ownership Model

We next turn to our analysis of how heterogeneity in the sample of firms—which exchange they choose to list with and whether they are state-owned or privately held— affects the baseline result. First, we divide the sample into two sub-samples: firms listed on the Shanghai exchange and those listed on the Shenzhen exchange. The results of analysis of the baseline regression for firms listed on the Shanghai exchange are presented in Table 6; for firms listed on the Shenzhen exchange, the results are presented in Table 7. As reported in Table 6, firms listed on the Shanghai exchange show no statistically significant relationship between R&D investment and firm performance outcomes. This is in sharp contrast to the results reported in Table 7, which illustrate that for firms listed on the Shenzhen exchange, there is a highly statistically significant positive impact of R&D on all measures of firm performance. Taken together, the results confirm hypothesis five, H5. Traditional manufacturing firms, which tend to list on the Shanghai exchange, do not stand to gain as much from R&D investment as do the high-tech firms listed on the Shenzhen exchange. For larger, more established, traditional manufacturing firms, product differentiation may not be as important in maintaining a firm's competitive edge as other strategies such as branding and marketing. However, for high-tech firms, differentiating products and services, delivering appropriate products to the appropriate consumer group,

seems important. For the high-tech firms listed on the Shenzhen exchange, R&D investment in product differentiation yields statistically significantly more market power, larger profits, and higher quality of earnings.

**Table 6.** Shanghai exchange listed firms.

| | Dependent Variable | | | |
|---|---|---|---|---|
| | Market Shares$_{t+1}$ | ROA$_{t+2}$ | ROA$_{t+1}$ | NETMARG$_{t+1}$ |
| | (1) | (2) | (3) | (4) |
| RDRATIO | 0.288 * | 0.356 | 0.522 | 1.217 |
| | (0.171) | (0.321) | (0.324) | (1.113) |
| TOP | 0.057 *** | 0.080 *** | 0.104 *** | 0.195 ** |
| | (0.012) | (0.023) | (0.023) | (0.079) |
| LIAB | 0.006 ** | −0.010 ** | −0.016 *** | −0.038 ** |
| | (0.002) | (0.004) | (0.005) | (0.016) |
| DUAL | −0.044 | −0.815 | −0.406 | −3.601 ** |
| | (0.279) | (0.519) | (0.520) | (1.787) |
| SHORTLIAB | −0.009 | −0.030 ** | −0.024 * | −0.127 *** |
| | (0.007) | (0.014) | (0.014) | (0.048) |
| RDRATIO*TOP | −0.005 | −0.004 | −0.008 | −0.015 |
| | (0.005) | (0.009) | (0.009) | (0.032) |
| Constant | 2.068 * | 5.018 *** | 4.047 *** | 8.317 * |
| | (1.213) | (1.442) | (1.448) | (4.974) |
| Industry Control within Manufacturing Sector | Y | N | N | N |
| Year Control | Y | Y | Y | Y |
| Observations | 2076 | 1557 | 2076 | 2075 |
| R$^2$ | 0.515 | 0.028 | 0.028 | 0.014 |
| Adjusted R$^2$ | 0.506 | 0.023 | 0.024 | 0.010 |
| Residual Std. Error | 4.853 (df = 2038) | 8.169 (df = 1548) | 9.450 (df = 2066) | 32.457 (df = 2065) |
| F Statistic | 58.507 *** (df = 37; 2038) | 5.649 *** (df = 8; 1548) | 6.667 *** (df = 9; 2066) | 3.352 *** (df = 9; 2065) |

Note: ***, **, and * denote the statistical significance at the 1%, 5%, and 10%, standard errors are shown in parentheses.

**Table 7.** Shenzhen exchange listed firms.

| | Dependent Variable | | | |
|---|---|---|---|---|
| | Market Shares$_{t+1}$ | ROA$_{t+2}$ | ROA$_{t+1}$ | NETMARG$_{t+1}$ |
| | (1) | (2) | (3) | (4) |
| RDRATIO | 0.272 *** | 0.937 *** | 0.950 *** | 10.678 ** |
| | (0.096) | (0.332) | (0.282) | (5.238) |
| TOP | 0.044 *** | 0.119 *** | 0.120 *** | 1.002 ** |
| | (0.008) | (0.030) | (0.024) | (0.450) |
| LIAB | 0.030 *** | −0.066 *** | −0.070 *** | −0.568 *** |
| | (0.003) | (0.011) | (0.009) | (0.173) |
| DUAL | 0.348 ** | −0.826 | −0.792 * | 5.289 |
| | (0.141) | (0.508) | (0.420) | (7.811) |
| SHORTLIAB | −0.008 * | 0.009 | 0.006 | 0.450 * |
| | (0.004) | (0.015) | (0.013) | (0.234) |
| RDRATIO*TOP | −0.007 *** | −0.015 | −0.015 * | −0.241 |
| | (0.003) | (0.010) | (0.008) | (0.152) |
| Constant | 1.313 ** | 1.518 | 2.373 | −52.446 * |
| | (0.623) | (1.756) | (1.460) | (27.157) |
| Industry Control within Manufacturing Sector | Y | N | N | N |
| Year Control | Y | Y | Y | Y |
| Observations | 2980 | 2235 | 2980 | 2980 |
| R$^2$ | 0.498 | 0.039 | 0.044 | 0.010 |
| Adjusted R$^2$ | 0.492 | 0.036 | 0.042 | 0.007 |
| Residual Std. Error | 3.373 (df = 2942) | 10.647 (df = 2226) | 10.172 (df = 2970) | 189.174 (df = 2970) |
| F Statistic | 79.018 *** (df = 37; 2942) | 11.367 *** (df = 8; 2226) | 15.334 *** (df = 9; 2970) | 3.314 *** (df = 9; 2970) |

Note: ***, **, and * denote the statistical significance at the 1%, 5%, and 10%, standard errors are shown in parentheses.

Next, we divide the sample into sub-samples based on whether the firm in question is state-owned or privately held. Table 8 reports the analysis of state-owned firms, while Table 9 reports the analysis of privately held firms. The results suggest there are significant differences in the relationship between R&D and firm performance outcomes across different kinds of firm ownership. As demonstrated in Table 8, coefficient estimates on R&D are all statistically insignificantly different from zero for all firm performance outcome variables for state-owned firms. In contrast, as reported in Table 9, for private firms, all R&D coefficient estimates are positive and nearly all are highly statistically significant. Taken together, the results confirm hypothesis six, H6. Our interpretation of these findings is that the incentives at SOEs make managers at SOEs too risk averse to invest in R&D in sufficient scale and/or in a way that enables them to reap the benefits in terms of firm performance outcomes. Privately held firms are better able to align managers' incentives to appropriate levels of risk, so privately held firms invest appropriately in R&D and enjoy the benefits that accrue in terms of firm performance outcomes. Careful interpretation of this result requires more analysis and may be the subject of a future study.

**Table 8.** State-owned firms.

| | **Dependent Variable** | | | |
| --- | --- | --- | --- | --- |
| | **Market Shares$_{t+1}$** | **ROA$_{t+2}$** | **ROA$_{t+1}$** | **NETMARG$_{t+1}$** |
| | **(1)** | **(2)** | **(3)** | **(4)** |
| RDRATIO | 0.301 | 0.512 | 0.503 | 1.439 |
| | (0.215) | (0.424) | (0.352) | (0.884) |
| TOP | 0.094 *** | 0.093 *** | 0.096 *** | 0.187 *** |
| | (0.015) | (0.030) | (0.024) | (0.060) |
| LIAB | 0.037 *** | −0.050 *** | −0.051 *** | −0.154 *** |
| | (0.006) | (0.012) | (0.010) | (0.024) |
| DUAL | 0.996 ** | −0.918 | −0.566 | −1.198 |
| | (0.391) | (0.782) | (0.653) | (1.638) |
| SHORTLIAB | −0.009 | −0.009 | −0.004 | 0.011 |
| | (0.009) | (0.018) | (0.015) | (0.037) |
| RDRATIO*TOP | −0.011 * | −0.012 | −0.013 | −0.034 |
| | (0.006) | (0.012) | (0.010) | (0.024) |
| Constant | −3.237 ** | 3.930 ** | 4.726 *** | 3.977 |
| | (1.398) | (1.983) | (1.628) | (4.085) |
| Industry Control within Manufacturing Sector | Y | N | N | N |
| Year Control | Y | Y | Y | Y |
| Observations | 1699 | 1278 | 1699 | 1699 |
| R$^2$ | 0.503 | 0.037 | 0.040 | 0.039 |
| Adjusted R$^2$ | 0.503 | 0.031 | 0.035 | 0.034 |
| Residual Std. Error | 4.797 (df = 1663) | 8.593 (df = 1269) | 8.229 (df = 1689) | 20.645 (df = 1689) |
| F Statistic | 50.157 *** (df = 35; 1663) | 6.158 *** (df = 8; 1269) | 7.882 *** (df = 9; 1689) | 7.540 *** (df = 9; 1689) |

Note: ***, **, and * denote the statistical significance at the 1%, 5%, and 10%, standard errors are shown in parentheses.

We suspect that the dominance of individual investors in China's financial markets may drive the results reported above in Table 5. Perhaps informational asymmetries lead individual investors to be more myopic in their investment decisions, focusing too much on current earnings and ignoring important information on firms' future growth potential. To further investigate this idea, we explore the relationship between firm R&D investment and the share of mutual fund ownership in the firms in our sample. Table 10 reports the results, demonstrating that higher R&D investment by firms attracts mutual fund investment, leading to a higher share of mutual fund ownership in the firm. This suggests that the more sophisticated information and analysis available to institutional investors leads them to consider reported earnings as a dynamic variable, and to make investment decisions based on more diverse, longer-term information about the firms, including R&D investment. Thus, the results reported in Table 10 confirm hypothesis seven, H7.

**Table 9.** Privately held firms.

| | Dependent Variable | | | |
|---|---|---|---|---|
| | **Market Shares$_{t+1}$** | **ROA$_{t+2}$** | **ROA$_{t+1}$** | **NETMARG$_{t+1}$** |
| | **(1)** | **(2)** | **(3)** | **(4)** |
| RDRATIO | 0.179 ** | 0.746 *** | 0.872 *** | 8.334 * |
| | (0.086) | (0.287) | (0.264) | (4.465) |
| TOP | 0.025 *** | 0.113 *** | 0.127 *** | 0.792 ** |
| | (0.007) | (0.025) | (0.023) | (0.380) |
| LIAB | 0.008 *** | −0.015 *** | −0.023 *** | −0.157 * |
| | (0.002) | (0.005) | (0.005) | (0.084) |
| DUAL | 0.086 | −1.152 *** | −0.924 ** | 2.953 |
| | (0.128) | (0.444) | (0.398) | (6.720) |
| SHORTLIAB | −0.003 | −0.003 | −0.002 | 0.335 * |
| | (0.004) | (0.013) | (0.012) | (0.202) |
| RDRATIO*TOP | −0.003 | −0.009 | −0.011 | −0.172 |
| | (0.003) | (0.009) | (0.008) | (0.132) |
| Constant | 3.013 *** | 1.271 | 0.960 | −52.679 ** |
| | (0.555) | (1.435) | (1.299) | (21.935) |
| Industry Control within Manufacturing Sector | Y | N | N | N |
| Year Control | Y | Y | Y | Y |
| Observations | 3357 | 2514 | 3357 | 3356 |
| R$^2$ | 0.567 | 0.030 | 0.034 | 0.006 |
| Adjusted R$^2$ | 0.562 | 0.027 | 0.031 | 0.003 |
| Residual Std. Error | 3.377 (df = 3319) | 10.273 (df = 2505) | 10.654 (df = 3347) | 179.815 (df = 3346) |
| F Statistic | 117.493 *** (df = 37; 3319) | 9.608 *** (df = 8; 2505) | 13.051 *** (df = 9; 3347) | 2.252 ** (df = 9; 3346) |

Note: ***, **, and * denote the statistical significance at the 1%, 5%, and 10%, standard errors are shown in parentheses.

**Table 10.** Research expense and institutional investors.

| | Dependent Variable | |
|---|---|---|
| | **INST$_{t+1}$** | |
| | **(1)** | **(2)** |
| RDRATIO | 0.606 *** | 0.591 *** |
| | (0.106) | (0.105) |
| TOP | 0.016 * | 0.018 ** |
| | (0.008) | (0.008) |
| LIAB | −0.003 | −0.003 |
| | (0.002) | (0.002) |
| DUAL | 0.375 ** | 0.382 ** |
| | (0.164) | (0.162) |
| SHORTLIAB | −0.016 *** | −0.014 *** |
| | (0.005) | (0.005) |
| RDRATIO*TOP | −0.004 | −0.004 |
| | (0.003) | (0.003) |
| Constant | 2.776 *** | 2.112 *** |
| | (0.493) | (0.504) |
| Year Control | N | Y |
| Observations | 5056 | 5056 |
| R$^2$ | 0.030 | 0.049 |
| Adjusted R$^2$ | 0.029 | 0.047 |
| Residual Std.Error | 4.965 (df = 5049) | 4.918 (df = 5046) |
| F Statistic | 26.252 *** (df = 6; 5049) | 28.878 *** (df = 9; 5046) |

Note: ***, **, and * denote the statistical significance at the 1%, 5%, and 10%, standard errors are shown in parentheses.

### 4.5. Summary of Results and Findings

Table 11 summarizes the results discussed above and provides some analysis of the implications of the findings. For clarity, the first two columns of Table 11 briefly repeat the hypothesis developed above in Section 2 from the existing theoretical and empirical

literature. Columns 3 and 4 link the hypothesis to a specific empirical model as developed in Section 3 and the table reporting the results of the empirical analysis above in this section. Column 5 reports whether the empirical analysis is carried out on the full sample of firms or, as in the case of some of the tests for heterogeneity across firms, a sub-sample of specific firms. Finally, column 6 reports whether the empirical results supported or refuted the stated hypothesis column 7 briefly discusses the implications of those results.

**Table 11.** Summary of findings and implications.

| | Hypotheses | Model | Table | Sample | Validation | Results and Implications |
|---|---|---|---|---|---|---|
| H1. | R&D and Market Power | Equation (1) | Table 3 | Full | Supported | R&D positively contributes to the firm gaining market power through more diversified products and services. |
| H2. | R&D and Profitability | Equation (2) | Table 3 | Full | Supported | R&D positively contributes to the firm's profitability. Larger R&D investments relative to the firm's competitors increases firm profits. |
| H3. | R&D and Earning Quality | Equation (3) | Table 3 | Full | Supported | R&D improves firm earnings quality. More diversified services and products as a result of higher R&D increases current and future revenue. |
| H4 | R&D and Shareholder Value | Equation (4) | Table 5 | Full | Refuted | The R&D investment's short-term effect on a firm's share price from the stock market is negative. Even though such a negative effect quickly disappears, it may negatively affect the managers' incentive to make R&D investment decisions. |
| H5. | Heterogeneity: Listing Location | Equations (1)–(3) | Tables 6 and 7 | Sub | Supported | More high-tech firms enjoy the return from R&D investments but not the traditional large-size manufacturing firms in the Chinese market. |
| H6. | Heterogeneity: SOEs vs. Privately-held Firms' | Equations (1)–(3) | Tables 8 and 9 | Sub | Supported | Privately held firms (that are not state owned) enjoy the economic benefits of higher R&D investments documented above, but state-owned enterprises do not reap any significant economic benefits from R&D expenditures. |
| H7. | R&D and Institutional Ownership Share | Equation (5) | Table 10 | Full | Supported | Firms with higher R&D investments attract a higher share of institutional mutual fund investors. |

All three baseline hypotheses are supported. Firms that have relatively high levels of R&D investment relative to other firms or over the time period studied command a higher share of the market (H1) and report higher returns on assets (H2) and net profit margins (H3).

Surprisingly, those benefits from R&D do not translate into higher firm valuation, refuting hypothesis four, H4. This may be partly due to heterogeneity in the baseline results across firms. Firms listed on the Shenzhen exchange, which tend to be high-tech firms, and firms that are privately held, reap the benefits of R&D investment, but firms listed on the Shenzhen exchange and state-owned enterprises do not (H5, H6). However, we interpret the lack of evidence that R&D investment translates into higher shareholder value as evidence that, as has been found in developed economies, some market participants—particularly individual investors which dominate China's financial markets—may underestimate the

benefits of R&D. As indirect evidence of this, we find that firms with higher levels of R&D investment attract relatively more institutional investors.

## 5. Conclusions and Policy Recommendations

### 5.1. Conclusions

In developed economies, product differentiation is an established business strategy for maintaining a firm's position in a competitive industry. In this paper, we argue that the investment in R&D to differentiate products and services in emerging markets brings even higher rewards in emerging markets. Using Chinese manufacturing industry data, we demonstrate that firms with higher R&D expand market power, boost profitability, and escalate earning quality.

There is some heterogeneity in the baseline result, however. The benefits of R&D investment are particularly high for high-tech firms that tend to list on the Shenzhen exchange, but are not enjoyed by more established, traditional industries within the manufacturing sector that tend to gravitate towards the Shanghai exchange. While private firms are able to leverage R&D into increased market power, profitability, and earnings quality, state-owned firms do not demonstrate any statistically significant improvements in firm performance from investments in R&D. Finally, while management's R&D investments significantly influence institutional investors' investment decisions in the Chinese manufacturing industry, individual investors are not able to fully recognize the benefits. Since individual investors still dominate China's financial markets, the overall market reaction to R&D does not capture the full potential benefits R&D offers to firms.

### 5.2. Policy Recommendations

These findings hold several implications for policymakers. We suggest priority be given to policies that encourage innovation and R&D in general, but also particularly at state-owned enterprises. Educate investors as to the kinds of R&D investments firms are pursuing and their potential benefits will lead to market incentives that reinforce such policies.

In general, policy should encourage R&D investments and reward innovation. R&D investment leads to more product differentiation and technological innovation. Policy incentives that encourage firms to invest in R&D bring benefits not only to the firm, but to broader society in the form of technological advancement, labor force training and experience, diversified product availability for consumers and overall economic efficiency.

Secondly, policymakers should find ways to promote R&D investments at state-owned enterprises. As our analysis demonstrates, currently, SOEs are not reaping the economic benefits that accrue to private firms that invest in R&D. This may be because SOEs are not investing in R&D at sufficient scale to see benefits in terms of firm performance outcomes. One feature of corporate governance at SOEs that may discourage investment in R&D is asymmetric loss functions in the case of project failure. This can be exacerbated by multiple levels of agency problems. A potential policy solution may be to encourage the SOEs to establish innovation funds that provide capital to local, smaller firms to invest in R&D. These funds could be monitored by a third-party and periodically reassessed.

Finally, there is a need for better investor education and transparency in reporting. Requiring firms to more completely and transparently disclose their R&D projects would educate the public, including individual investors, as to the potential gains from R&D. This would create a virtuous cycle of attracting more investors, triggering a more robust market reaction and thus providing more incentive for firms to invest in R&D. Market forces will reinforce policies to promote investment in R&D.

### 5.3. Limitations and Directions for Future Research

The results also suggest several interesting directions for future research. One interesting and potentially important factor not explored in this study is the stage of the R&D projects undertaken by those firms that invest in R&D. The stage of the research

and development projects may influence the outcome variables examined in this study. Investors' expectations around R&D projects that have just been initiated by a given firm may be very different from expectations formed around an R&D project reaching maturity. The economic impact on market power, firm profitability, and earnings quality may only manifest when the R&D project is successful or when the newly developed technology is actually used and applied. Detailed information on specific R&D projects, including some measure of the stage of the project's development, would provide more accurate and interesting analysis.

**Author Contributions:** Conceptualization—D.S. and H.M.; methodology—D.S. and H.M.; resources—D.S. and H.M.; writing—original draft—D.S. and H.M.; writing—review and editing—D.S. and H.M.; visualization—D.S. and H.M. All authors have read and agreed to the published version of the manuscript.

**Funding:** This research received no external funding.

**Data Availability Statement:** Data are available at Choice Eastmoney and CSMAR upon subscription.

**Conflicts of Interest:** The authors declare no conflict of interest.

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
