# Peer review of "Should Firms in Emerging Markets Invest in R&D? Evidence from China’s Manufacturing Sector"

_jrfm, doi:10.3390/jrfm15110517_

Round 1

Reviewer 1 Report

The paper is interesting, it deals with an important topic of the investment in R&D in a very important emerging market, and it is my pleasure to review it. The paper has merits, it is detailed, well-organized, and uses a solid scientific and logical tool. Methodology and approaches are interesting, systematic and comprehensive.

However, I would have some considerations and suggestions for improving the quality of the article.

The H6 Hypotheses In the relatively short-term, returns to shareholders are negatively impacted by R&D investment, since the earning per share would be relatively low, but mutual fund share ownership increases with R&D investment, is complicated and equivocal, with many conditionalities and alternatives which dilutes its meaning. It is unclear what its validation or invalidation would mean.

The authors should define more clearly the expression(s) Higher investments in R&D/higher profit etc. … higher than what? A benchmark, a term of comparison (across sectors, by nature of ownership, over time or across countries) could clarify this.

The paper proposes 6 hypotheses. Although they are analysed and discussed in Chapter 4, following them in the text (and understanding the connection between them) is quite difficult. We recommend a paragraph or a table that clearly summarizes the result of the analysis (their validation/invalidation) and the resulting consequences.

Implicitly, in the Final Conclusions, we recommend a more careful reiteration of the research’s context and results, and, where appropriate, several policy recommendations. Mentioning the limits of the research and, if the case, the topics that can be deepened in future researches (a kind of invitation to an academic debate on the insufficiently clarified aspects) would improve the usefulness and relevance of the research.

Minor formal issues

The paper must be formally reviewed, there are small typos, different fonts and spacing from one paragraph to another, etc.

Thank you for the opportunity to review this article and good luck!

Author Response

Dear Reviewer,

Thank you so much for helping us improve the paper's quality. Please see the attached word document for a point-to-point update.

Reviewer 2 Report

The work is clearly described and the methods used are well described and justified. So I have only one important note. On line 215, you write "descriptive statistics reported in Table 2", but you have descriptive statistics in Table 1.

Author Response

(The authors gave the same response as above.)

Round 2

Reviewer 1 Report

The authors carefully addressed the suggestion and recommendations made. Therefore, in its revised form, the paper is suitable for publication.